# Evaluating operational parameters of the *care*HPV, GeneXpert, AmpFire, and MA-6000 HPV systems for cervical precancer screening: Experience from Battor, Ghana

Kofi Effah[1], Comfort Mawusi Wormenor[1], Ethel Tekpor[1], Joseph Emmanuel Amuah[1,2], Nana Owusu M. Essel[3]*, Isaac Gedzah[1], Seyram Kemawor[1], Benjamin Tetteh Hansen[1], Bernard Hayford Atuguba[1], Gifty Belinda Klutsey[1], Edna Sesenu[1], Stephen Danyo[1], Patrick Kafui Akakpo[4]

1 Catholic Hospital, Battor, via Sogakope, Volta Region, Ghana, 2 Faculty of Medicine, School of Epidemiology and Public Health, University of Ottawa, Ottawa, ON, Canada, 3 Faculty of Medicine and Dentistry, Department of Emergency Medicine, College of Health Sciences, University of Alberta, Edmonton, AB, Canada, 4 Department of Pathology, University of Cape Coast, School of Medical Sciences, Clinical Teaching Center, Cape Coast, Ghana

* nanaowus@ualberta.ca

**Data Availability Statement:** All relevant information are contained within the paper. The

## Abstract

In response to calls by the World Health Organization for cervical precancer screening services in low-resource settings to lean toward HPV DNA testing, a number of testing platforms have been made available. This study aimed to evaluate the operational parameters of four HPV testing systems in previous (*care*HPV) and current (GeneXpert, AmpFire, and MA-6000) use in a secondary healthcare setting in terms of 'appropriateness', ease of use, throughput, and diagnostic yield. This descriptive retrospective cohort analysis included 6056 women who presented to our facility between June 2016 and March 2022 for cervical precancer screening via HPV testing. A large majority of this cohort underwent AmpFire testing (55.8%), followed by *care*HPV (23.3%), MA-6000 (14.7%), and GeneXpert (6.1%). MA-6000 showed the highest hr-HPV positivity rate of 26.4% (95% CI, 23.6–29.5), followed by AmpFire (17.2%; 95% CI, 15.9–17.5). GeneXpert and *care*HPV showed similar hr-HPV positivity rates of 14.8% (95% CI, 11.3–18.8) and 14.8% (95% CI, 13.0–16.8), respectively. For the AmpFire and MA-6000 platforms, which utilize similar detection and reporting formats, we found a significant excess detection rate of 9.2% (95% CI, 6.1–12.4; *p*-value <0.0001) for MA-6000 compared to AmpFire. At the genotype level, MA-6000 also detected significantly higher rates of HPV 16 and *other* hr-HPV types (both *p*-values <0.001) than AmpFire; there was no difference in detection for HPV 18. Based on our experiences and preliminary analysis, we believe that the choice of HPV testing platform cannot be accomplished with a one-size-fits-all approach. Factors worth considering are the financial implications of platform acquisition, costs to clients, and throughput when screening programs are not sufficiently large. We describe our successes and challenges with the different platforms which we believe will be helpful to centers in low-income countries as they transition into using HPV DNA testing for cervical precancer screening.

datasets can be accessed at: https://doi.org/10.6084/m9.figshare.21801307.v1.

**Funding:** The authors received no specific funding for this work.

**Competing interests:** The authors have declared that no competing interests exist.

## Introduction

Cervical cancer is the fourth commonest malignancy among women worldwide and constitutes a significant public health problem, with an estimated 85% of incident cases occurring in the developing world [1]. In Ghana, it is the second most prevalent cancer (with breast cancer taking the lead) and cause of cancer-related mortality [2]. Similar to other sub-Saharan countries, an estimated 3052 new cases and 1556 deaths result from cervical cancer annually [3]. Data on the prevalence and genetic distribution of HPV infection among women in Ghana are limited. Therefore, it is quite challenging to diagnose HPV infection and identify its associated clinical factors in Ghana, being limited by the low level of awareness among the general female population, centralization of screening services, and negative sociocultural and religious beliefs [4]. For a country that has no clinical guidelines pertaining to cervical precancer screening and HPV management, it is essential to gather country-specific data to serve as a basis for policy-making and future research.

A variety of molecular methods exist for the detection of oncogenic HPV and it is generally accepted that persistent positivity is essential for cervical carcinogenesis [5]. HPV DNA assay methods are used as primary screening tests, often followed by other methods such as cytology for triage [6]. In resource-limited settings where Papanicolaou/cytologic test implementation has been unsuccessful, the World Health Organization (WHO) recommends HPV DNA testing as a substitute [7]. Notable among these tests are the *care*HPV (Qiagen GmBH, Hilden, Germany); the GeneXpert HPV test (Cepheid, Sunnyvale, CA, USA); the AmpFire HPV system (Atila BioSystems, Inc., Mountain View, USA); and MA-6000 (Sansure Biotech Inc., Hunan, China), which were introduced at our center, the Cervical Cancer Prevention and Training Center (CCPTC), Battor, Ghana and are being considered in this study.

In response to efforts made to manufacture tests that are inexpensive and suitable for low-resource settings, require minimal expertise, and are effective at the point of care, tests such as *care*HPV have been made available [8]. It is a semi-rapid qualitative test and a simplified version of the Digene Hybrid Capture II technology [9]. *care*HPV shows positive results in the presence of the DNA of any of 14 hr-HPV subtypes (13, and/or 16, 18, 31, 33, 35, 39, 45, 52, 56, 58, 59, 66, 68) [10]. In addition to being easily implementable, economically sustainable, and having a high acceptance rate among users, there is little need for samples to be refrigerated and it can use a battery-operated machine [9]. In a recent meta-analysis [11], *care*HPV showed adequate sensitivity (92.3%; 95% confidence interval [CI], 81.0–97.1) for detecting cervical intraepithelial neoplasia (CIN) grades 2+. The platform has also been validated against an internationally-accepted reference standard [12,13]. *care*HPV is no longer used at the CCPTC due to its inability to differentiate between subtypes and the need to batch samples before testing, which delays testing.

The GeneXpert HPV test is a qualitative *in vitro* real-time polymerase chain reaction (PCR) assay that identifies 13 known hr-HPV subtypes (16, 18, 31, 33, 35, 39, 45, 51, 52, 56, 58, 59, and 68) and a possibly high-risk type (HPV 66) [14]. Notably, the GeneXpert HPV cartridge test yields results in about 1 h, thereby permitting screening within a short time, early diagnosis, and treatment. While different GeneXpert platforms have different throughputs, the one in use at Battor has low throughput, running a maximum of 4 samples at a time. Widely used in the detection of tuberculosis and its associated drug resistance, the availability of this platform in many rural hospitals on the National Tuberculosis Programme constitutes a unique opportunity for its use as part of a national HPV-based cervical precancer screening program.

The AmpFire HPV system utilizes a simple molecular detection mechanism. Instead of DNA extraction, it amplifies DNA isothermally and detects all 15 hr-HPV subtypes in a single tube sample. This amplification process is also relatively fast (reaction time, 40 min). Thus, the time from sample to answer is within 2 h (including a 10-min heating period). Its simplicity

and speed make it a great fit for resource-limited areas [15]. At the CCPTC, Battor, the Amp-Fire technology has been available since June 2019.

A newer hr-HPV DNA testing platform, MA-6000, allows for multiple tests to be run, including hepatitis B, HIV, and COVID-19 tests, just like the GeneXpert and AmpFire platforms. The MA-6000 platform utilizes a real-time quantitative PCR cycler. MA-6000 classifies hr-HPV types qualitatively in a manner similar to AmpFire. In addition, it is associated with increased throughput and allows up to 96 samples to be processed simultaneously, just like the AmpFire platform. The MA-6000 platform was introduced at our center in September 2021.

While the performances of *care*HPV, GeneXpert, AmpFire, and visual inspection methods have been compared in quite a number of low-income settings for the assessment of cytologically diagnosed lesions and malignancies, a four-way comparison of the operational parameters of these three platforms in addition to MA-6000 has not been performed. Thus, the present study aimed to compare the operational features of these four hr-HPV testing systems in current and previous use in a secondary healthcare setting in Ghana for primary cervical precancer screening in terms of 'appropriateness', ease of use, throughput, and diagnostic yield.

## Materials and methods

### Study design, setting, and data collection

This retrospective descriptive cross-sectional study evaluated the data for all women who underwent HPV DNA screening between June 1, 2016 and March 31, 2022 at the CCPTC, Battor, Ghana. Data collected included details of women's sociodemographic characteristics (including age, marital status, number of children), previous treatment(s), smoking status, contraceptive use, screening method, HPV tests performed, and HPV DNA test results.

The CCPTC is situated within the Catholic Hospital, Battor, Volta Region, Ghana, and was formally opened in May 2017. Using a structured modular program, the center aims to build capacity in cervical cancer prevention in Ghana and beyond, primarily by training health workers in practical competences involved in taking samples for HPV testing, visual inspection with acetic acid (VIA), colposcopy, and setting up cervical screening services at their institutions. The CCPTC sees women from all over Ghana and neighboring countries who patronize its services for cervical precancer screening, treatment, and follow-up. At the community level and for specific vulnerable groups, the CCPTC conducts community outreach programs to create awareness about cervical cancer and offers women to opportunity to be screened.

Routinely at the CCPTC, mobile colposcopy using the Enhanced Visual Assessment (EVA) platform or VIA is performed simultaneously with cervicovaginal sampling for HPV DNA testing using the *care*HPV, GeneXpert, AmpFire, or MA-6000 platforms. EVA or VIA is also sometimes performed as a primary screening procedure for cervical precancer. Women who test positive for hr-HPV types on standalone testing also undergo EVA or VIA.

### Ethical considerations

Verbal informed consent was obtained from all women prior to screening. The Research Ethics Committee of the Catholic Hospital, Battor, granted ethical approval for this study (approval no. CHB-ERC-002/07/19). All data were de-identified after data extraction prior to performing the analysis.

### Cervicovaginal sample collection and storage

At the screening visit, each woman received counseling about the advantages of screening, as well as its associated risks and potential outcomes. After obtaining verbal informed consent, a

speculum was inserted after placing the women in the dorsal lithotomy position to reveal the cervix, allowing for the collection of cervical samples. For health worker-collected samples, the choice of whether dry brushes or ThinPrep would be used depended on what the women wanted after discussions with them. Health workers used cytobrushes (more often) or cotton-tipped applicators depending on the availability. Women who wanted cytology in addition to HPV DNA testing had samples put in ThinPrep to be able to run both tests. For GeneXpert, we used PreservCyt (ThinPrep) which was usually a health worker collected sample (with a Cervex brush) that was rinsed in ThinPrep. For AmpFire and MA-6000, we used both dry brush samples and ThinPrep, with the samples taken by health workers, while *care*HPV samples were placed in *care*HPV collection medium. Samples for GeneXpert were usually run the same day. For AmpFire and MA 6000, the samples were usually run within two weeks. *care*HPV took much longer (sometimes up to 6 weeks) because samples needed to be batched. Women who opted for self-sampling were taught how to use the Evalyn self-sample brush (Rovers Medical Devices B.V., Oss, Netherlands). After taking the samples, the Evalyn brushes were sealed, kept in a dry space at room temperature, and sent to the main laboratory for analysis within a week. Dry brush samples were stored in a freezer at -16˚C, while ThinPrep samples were stored in an airconditioned room at a temperature range of 16–20˚C.

Because each of the HPV DNA test platforms were introduced at different times at the CCPTC, the women generally had the option of choosing which test they wanted done, after giving them details, including how long it would take to get results and the cost involved. Some of the women who wanted their results the same day paid more for the GeneXpert instead of *care*HPV.

## Laboratory processing of cervicovaginal samples for HPV DNA assays

*care*HPV. *care*HPV tests were performed using kits according to the manufacturer's instructions [16]. Summarily, the specimens were mixed with lysis buffer to dissolve the cells. HPV DNA was then denatured to their single-strand forms by heating the lysate mixture; by so doing, the single-stranded DNA was hybridized with full-length complement RNA to yield HPV DNA/RNA hybrids. Monoclonal antibody-coated magnetic beads were then added, followed by alkaline phosphatase to act upon the resulting chromogenic substrate. The intensity of the light generated due to the reaction of the chromogenic substrate was indicative of the amount of HPV DNA contained in each specimen. A positive test result was diagnosed by calculating the ratio of relative light unit (RLU) to the mean value of the minimum positive control (1.0, standardized as a reading ≥0.5 pg/ml in the specimen). An RLU value below the cut-off value implies that the specimen contains insufficient or no hr-HPV DNA and is thus considered a negative test.

GeneXpert. GeneXpert HPV assays were also performed at the laboratory of the Catholic Hospital, Battor using aliquots of approximately 1.5 ml stored at ambient temperature prior to examination. All GeneXpert tests were performed in adherence to the manufacturer's instructions within 6 weeks of sample collection. Details regarding HPV GeneXpert testing have been described elsewhere [17]. In brief, the *E6* and *E7* genes of the hr-HPV types being targeted were amplified simultaneously using five fluorescent channels (for HPV types 16, 18/45, 31/33/35/52/58, 51/59, and 39/56/66/68). A sixth channel (HMBS) was used as a control to ensure specimen adequacy. Reports of assays were considered positive if any of the aforementioned HPV types were detected, with separate results for types 16 and 18/45.PV.

AmpFire. AmpFire assays were performed by centrifuging 1 ml of ThinPrep solution after which the resulting pellets of cells were treated with heat (after discarding the resulting

supernatant) in lysis buffer for 10 min without extracting DNA, according to the manufacturer's instruction [18]. Two microliters of the lysed sample were mixed with 23 μl of mastermix reaction solution for a total of 60 cycles in real-time fluorescence detection. A novel isothermal multiplex method of amplification coupled with real-time fluorescence detection (referred to as OMEGA amplification) was used to detect 15 hr-HPV types. This procedure is qualitatively designed to detect the following using four dye types: FAM (for hr-DNA), HEX as an internal control, CY5 (for HPV 16), and ROX (HPV 18).

**MA-6000.** MA-6000 testing was also performed in strict accordance with the manufacturer's instructions [19], details of which have been published elsewhere [20]. Briefly, a pure fraction of DNA was isolated in solution by adding the manufacturer's sample release reagent and incubating the mixture for 10 min at room temperature. Thereafter, PCR was initiated using 50 μl of the processed specimen and run on the MA-6000 device for a total of 45 cycles. Fluorescence data were then collected during amplification at 57˚C for 30 s. Test outputs were read and interpreted strictly according to the manufacturer's instructions. The MA-6000 kit is qualitatively designed to detect the following using four types of dye: FAM (for HPV 18); HEX (for detecting β-globin as an internal control); CY5 (for HPV 16); and ROX (for detecting HPV 31, 33, 35, 39, 45, 51, 52, 53, 56, 58, 59, 66, and 68 without differentiation).

### Key definitions

In evaluating the tests, we discuss 'appropriateness' subjectively as how suitable an HPV test platform is for implementation in a setting, encompassing the ability to meet the clinical needs of the screened population and adaptability to resource availability. We define ease of use as the simplicity and user-friendliness of the test platforms, including ease of sample collection, preparation, and handling, the complexity of the test procedure, and the need for specialized training. Throughput denotes the number of samples each platform can process within a given timeframe, as a measure of its capacity and efficiency, particularly in high-volume testing environments, while diagnostic yield refers to the ability of each HPV test platform to accurately identify hr-HPV in samples, as well as the ability to distinguish among types.

### Statistical analysis

We present descriptive statistics for all sociodemographic and clinical variables assessed in our study cohort. Categorical variables and prevalence estimates are described using frequencies and proportions, along with their binomial exact 95% confidence intervals (CIs). We also describe continuous variables using means with standard deviations for symmetrically distributed data or medians with interquartile ranges (IQRs) for data with asymmetrical distributions. Associations between categorical variables were assessed using Pearson's chi-squared test of independence and one-way analysis of variance was used to compare the means of symmetric continuous variables across more than two categories. The Kruskal–Wallis equality-of-populations rank test was used to compare medians across more than 2 populations for skewed continuous variables. All data cleaning and analyses were performed using Stata 15 (StataCorp LLC, College Station, TX, USA).

## Results

### Overall and platform-stratified sociodemographic and clinical details of the study cohort

Overall and platform-stratified details pertaining to the social, demographic, and clinical characteristics of the women are shown in Table 1. In total, 6056 women underwent HPV DNA

**Table 1. Sociodemographic and clinical details of women (n = 6056) who underwent cervical precancer screening via HPV DNA testing using the *care*HPV, GeneXpert, AmpFire, or MA-6000 platforms.**

| Characteristic | Overall | AmpFire | *care*HPV | GeneXpert | MA-6000 | *p*-value |
|---|---|---|---|---|---|---|
| Age, mean (SD) | 39.4 (9.5) | 39.6 (9.2) | 39.2 (9.4) | 39.7 (8.8) | 39.1 (10.8) | 0.2481$^\$$ |
| Marital status, n (%) | | | | | | <0.001 |
| Single | 903 (14.9) | 473 (14.0) | 218 (15.4) | 58 (15.6) | 154 (17.3) | |
| Has a steady partner | 1229 (20.3) | 735 (21.8) | 240 (17.0) | 59 (15.9) | 195 (21.8) | |
| Married | 3172 (52.4) | 1715 (50.8) | 786 (55.6) | 222 (59.7) | 449 (50.3) | |
| Divorced | 414 (6.8) | 246 (7.3) | 90 (6.4) | 24 (6.6) | 54 (6.1) | |
| Widowed | 290 (4.8) | 165 (4.9) | 75 (5.3) | 9 (2.4) | 41 (4.6) | |
| Missing | 48 (0.8) | 43 (1.3) | 5 (0.4) | 0 (0.0) | 0 (0.0) | |
| Number of children, median (IQR) | 1 (0, 2) | 1 (0, 2) | 1 (0, 2) | 1 (0, 3) | 1 (0, 3) | 0.0001* |
| Highest level of education, n (%) | | | | | | <0.001 |
| No formal education | 514 (8.9) | 385 (11.4) | 23 (1.6) | 11 (3.0) | 122 (13.7) | |
| Elementary education | 2166 (35.8) | 523 (15.5) | 1263 (89.3) | 257 (69.1) | 123 (13.8) | |
| Secondary education | 2099 (34.7) | 1569 (46.5) | 80 (5.7) | 50 (13.4) | 400 (44.8) | |
| Tertiary education | 1101 (18.2) | 793 (23.5) | 40 (2.8) | 52 (14.0) | 216 (24.2) | |
| Vocational/technical/other | 119 (2.0) | 79 (2.3) | 7 (0.5) | 2 (0.5) | 31 (3.5) | |
| Missing | 30 (0.5) | 28 (0.8) | 1 (0.1) | 0 (0.0) | 1 (0.1) | |
| Religious faith, n (%) | | | | | | <0.001 |
| Christian | 5571 (92.2) | 3150 (93.3) | 1401 (99.1) | 357 (96.0) | 835 (93.5) | |
| Islam | 266 (4.4) | 10 (0.3) | 0 (0.0) | 0 (0.0) | 0 (0.0) | |
| African traditional religion | 22 (0.4) | 16 (0.5) | 1 (0.1) | 0 (0.0) | 5 (0,6) | |
| Other | 8 (0.1) | 6 (0.2) | 0 (0.0) | 0 (0.0) | 2 (0.2) | |
| None | 6 (0.1) | 4 (0.1) | 1 (0.1) | 1 (0.3) | 0 (0.0) | |
| Missing | 183 (3.1) | 0 (0.0) | 1 (0.1) | 0 (0.0) | 0 (0.0) | |
| Smoker, n (%) | 29 (0.5) | 18 (0.5) | 9 (0.6) | 1 (0.3) | 1 (0.1) | <0.001 |
| HIV status, n (%) | | | | | | <0.001 |
| Positive | 133 (2.2) | 96 (2.8) | 2 (0.1) | 0 (0.0) | 35 (3.9) | |
| Negative | 2513 (41.5) | 2130 (63.1) | 57 (4.0) | 61 (16.4) | 265 (29.7) | |
| Unknown/missing | 3410 (56.3) | 1151 (34.1) | 1355 (95.8) | 311 (83.6) | 593 (66.4) | |
| Earns income, n (%) | | | | | | <0.001 |
| Yes | 4062 (67.1) | 3004 (89.0) | 158 (11.2) | 117 (31.5) | 783 (87.7) | |
| No | 476 (7.9) | 337 (10.0) | 19 (1.3) | 11 (3.0) | 109 (12.2) | |
| Missing | 1518 (25.1) | 36 (1.1) | 1237 (87.5) | 244 (65.6) | 1 (0.1) | |

hr-HPV, high-risk human papillomavirus; SD, standard deviation; IQR, interquartile range; CI, confidence interval.

$^\$$ One-way analysis of variance (ANOVA) F-test to compare means across four cohorts.

* Kruskal–Wallis equality-of-populations rank test to compare median parity across four cohorts.

testing using any platform during the study period. A large majority of the cohort underwent AmpFire testing (n = 3377, 55.8%), followed by *care*HPV (n = 1414, 23.3%), MA-6000 (n = 893, 14.7%), and GeneXpert (n = 372, 6.1%). At presentation, the mean age was 39.4 (standard deviation, 9.5) years, with a majority of the women being either married (52%) or having a steady sexual partner (20%). The median parity was 1 (IQR, 0–2) and a majority (67%) earned an income. A minority of the women (18%) had completed tertiary education, while 9% had no formal education, 36% had completed elementary education, 35% had completed secondary education, and 2% had completed vocational or technical training. A large majority of the women screened (92%) were Christians, 0.4% were African traditionalists, 4%

were Muslims, and 0.1% had other faiths. With respect to risk factors, a high majority of the women had never smoked (99.5%); 2% self-reported a positive HIV status, while 42% reported a negative HIV status, and the remaining had unknown statuses or missing data.

When disaggregated according to platform, we observed a similar age distribution among the four sub-cohorts of women tested across the platforms ($p$-value = 0.2481). On the other hand, the four platforms showed statistically significant differences in the distributions of religion ($p$-value <0.001), highest education level ($p$-value <0.001), whether or not a participant earned an income ($p$-value <0.001), number of children ($p$-value <0.001), smoking status ($p$-value <0.001), HIV status ($p$-value <0.001), and marital status ($p$-value <0.001) among women tested (Table 1).

## Overall and platform-stratified hr-HPV prevalence estimates of the study cohort

Table 2 presents the distributions of hr-HPV positivity and their respective rates among the study participants subjected to primary HPV screening with each of the platforms. Overall, 1080 out of the 6056 participants (17.8%; 95% CI, 16.9–18.8) tested hr-HPV-positive on any of the four platforms under study. In terms of detection rates, MA-6000 showed the highest hr-HPV positivity rate of 26.4% (95% CI, 23.6–29.5), followed by AmpFire (17.2%; 95% CI, 15.9–17.5). GeneXpert and *care*HPV showed similar hr-HPV positivity rates of 14.8% (95% CI, 11.3–18.8) and 14.8% (95% CI, 13.0–16.8), respectively.

Given that the AmpFire and MA-6000 platforms utilize similar detection and reporting formats, we statistically compared their detection rates, which showed a statistically significant excess overall hr-HPV detection rate of 9.2% (95% CI, 6.1–12.4; $p$-value <0.001) for MA-6000 compared to AmpFire. In terms of genotype distribution, MA-6000 again significantly detected 1.5% (95% CI, 0.4–2.7; $p$-value = 0.001) more cases of HPV 16 than AmpFire.

**Table 2. Overall and platform-stratified hr-HPV prevalence estimates and genotypes detected.**

| Testing platform | No. of participants (%) | No. positive | hr-HPV prevalence estimate (95% CI) |
|---|---|---|---|
| *care*HPV [a] | 1414 (23.3) | 210 | 14.8 (13.0–16.8) |
| GeneXpert | 372 (6.1) | 55<br>HPV 16, n = 8 (14.5%)<br>HPV 18/45, n = 8 (14.5%)<br>P3 [b], n = 27 (49.1%)<br>P4 [c], n = 7 (12.7%)<br>P5 [d], n = 7 (12.7%) | 14.8 (11.3–18.8) |
| AmpFire | 3377 (55.8) | 579<br>HPV 16, n = 45 (7.8%)<br>HPV 18, n = 73 (12.6%)<br>*Other* [e] hr-HPV type(s), n = 523 (90.3%) | 17.2 (15.9–17.5) |
| MA-6000 | 893 (14.7) | 236<br>HPV 16, n = 26 (11.0%)<br>HPV 18, n = 24 (7.6%)<br>*Other* [e] hr-HPV type(s), n = 208 (88.1%) | 26.4 (23.6–29.5) |
| Overall | 6056 (100.0) | 1080 | 17.8 (16.9–18.8) |

hr-HPV, high-risk human papillomavirus; CI, confidence interval.

[a] careHPV detects HPV 16, and/or 18, 31, 33, 35, 39, 45, 51, 52, 56, 58, 59, 66, 68 (without distinction).

[b] GeneXpert fluorescent channel 3 (P3) detects HPV 31 and/or 33, 35, 52, 58.

[c] GeneXpert fluorescent channel 4 (P4) detects HPV 51 and/or 59.

[d] GeneXpert fluorescent channel 5 (P5) detects HPV 39, and/or 56, 66, 68.

[e] Qualitative AmpFire and MA-6000 tests detect 13 hr-HPV types together (31, 33, 35, 39, 45, 51, 52,53, 56, 58, 59, 66, 68) as other hr-HPV types.

Although the detection rate of HPV 18 was slightly higher for MA-6000 (2.7% vs. 2.2%) than for AmpFire, the difference was not statistically significant (*p*-value = 0.348). The collective detection rate of *other* hr-HPV types (encompassing HPV 31, and/or 33, 35, 39, 45, 51, 52,53, 56, 58, 59, 66, 68, without distinction) was also significantly higher for MA-6000 than for AmpFire (23.3% vs. 15.5%; difference, 7.8; 95% CI, 4.8–10.8; *p*-value <0.001).

## Discussion

This study aimed to compare the operational parameters of four HPV-based primary screening platforms (*care*HPV, GeneXpert, AmpFire, and MA-6000) in current and previous use at our center in terms of 'appropriateness', ease of use, throughput, and diagnostic yield. Each platform was evaluated among a specific cohort of women to offer in-depth insights into their distinct characteristics and operational nuances. This work was deemed essential due to a paucity of evidence from the developing world despite recent calls by the WHO to adopt an HPV-based approach in low-resource settings [21]. Further, HPV testing has been found to predict the risk of cervical precancers and cancers with better accuracy and much sooner than cytology-based methods [22]Click or tap here to enter text. Therefore, apart from using HPV DNA testing to triage borderline cytologic lesions, HPV testing has proven to be accurate in primary cervical precancer screening alone [23] or as a co-test [24]. As far as we are aware, our center is one of a few settings with good experience with the use of all four HPV testing platforms in routine cervical precancer screening work. The overall positive detection rate of hr-HPV in our cohort (17.8%; 95% CI, 16.9–18.8) was nearly half (32.3%) that reported among women living in the North Tongu District, Ghana [25] and exceeded the rate of 10.7% among women at an outpatient gynecologic setting in Accra [26] and 13.9% among pregnant women in the Western region of Ghana [27], but was lower than the 47.6% reported among incarcerated women in a medium-security prison in Ghana [28]. Next, we summarize and compare our experiences with each platform under study (Table 3).

Each platform allows relatively simple and quick HPV DNA testing without a need for separate DNA extraction procedures. GeneXpert, AmpFire, and MA-6000 allow the use of liquid-based media such as PreservCyt (ThinPrep). This makes it possible to perform liquid-based cytology with HPV DNA testing (or as a reflex for positive HPV DNA tests). On the other hand, *care*HPV comes with its own collection medium which does not allow for liquid-based cytology.

It is difficult to have a head-to-head comparison of costs because different arrangements were made to acquire the HPV testing platforms at Battor. For *care*HPV, crowdfunding was used to raise the USD 13,500 agreed on by the representative of Qiagen in West Africa [29]. For GeneXpert, we hitchhiked on the National Tuberculosis Programme which has over 100 platforms in hospitals across Ghana, and so did not have to pay for the platform; we only had to purchase cartridges for HPV DNA testing. The AmpFire platform was purchased by the Member of Parliament for North Tongu District, in which the Catholic Hospital, Battor is situated [30]. Our hospital acquired the MA-6000 platform as a gift from mPharma as part of the mPharma 10,000 Women Campaign, which aimed to provide 10,000 women in Ghana and Nigeria with free cervical precancer screening via HPV DNA testing [31].

Again, despite utilizing a similar style and format of reporting hr-HPV types, AmpFire and MA-6000 showed a statistically significant difference in detection rates. While the reason for this difference is unclear, the method of sampling has been found to affect HPV detection in several ways. First, self-sampling might not actually sample cervical tissue, but rather the vaginal epithelium, increasing the likelihood of HPV detection due to the larger area of potentially HPV-infected cells compared to the cervix [32]. Interestingly, significantly more women had

**Table 3. Summary of operational parameters of HPV testing with the *care*HPV, GeneXpert, AmpFire, and MA-6000 platforms at the CCPTC, Battor.**

| Platform | *care*HPV | GeneXpert | AmpFire | MA-6000 |
|---|---|---|---|---|
| **Manufacturer** | Qiagen GmBH, Hilden, Germany | Cepheid, Sunnyvale, CA, USA | Atila BioSystems, Inc., Mountain View, CA, USA | Sansure Biotech Inc., Hunan, China |
| **HPV types detected** | HPV 16, 18, 31, 33, 35, 39, 45, 51, 52, 56, 58, 59, 66, and 68 (not individually) | - HPV 16, 18/45 (specific identification)<br>- 11 other hr-HPV type(s): 31, 33, 35, 39, 51, 52, 56, 58, 59, 66, 68 | - HPV 16, 18 (specific identification)<br>- 13 hr-HPV types together (31, 33, 35, 39, 45, 51, 52,53, 56, 58, 59, 66, 68)<br>- Full genotyping possible (but more expensive with lower throughput) | - HPV 16, 18 (specific identification)<br>- 13 hr-HPV types together (31, 33, 35, 39, 45, 51, 52,53, 56,58, 59, 66, 68)<br>- Full genotyping possible (but more expensive with lower throughput) |
| **Sample type** | Cervical specimen in *care*HPV collection medium | Cervical specimen in PreservCyt/ ThinPrep liquid cytology specimens | Genital swabs (dry brushes, swabs, or PreservCyt/ThinPrep) | Genital swabs (dry brushes, swabs, or PreservCyt/ThinPrep) |
| **Batching of samples** | Yes (up to 94 samples with positive and negative controls) | Singly or batches of up to 2, 4 [a], 16, 48, or 80 depending on the module | Singly or batches of up to 94 samples with positive and negative controls | Singly or batches of up to 94 samples with positive and negative controls |
| **Duration of test** | About 2.5 h including hands-on time | 58 min | Less than 2 h for up to 94 samples including hands-on time | Less than 2 h for up to 94 samples including hands-on time |
| **8 h throughput** | Up to 270 | 8, 16, 32, 128, 384, 640 [b] | Up to 376 | Up to 376 |
| **Number of tests (done in Battor)** | 1414 | 372 | 3377 | 893 |
| **Number of positives** | 210 | 55 | 579 | 236 |
| **Proportion of hr-HPV positives** | 14.8 | 14.8 | 17.2 | 26.4 |

hr-HPV, high-risk human papillomavirus; CCPTC, Cervical Cancer Prevention and Training Centre.

[a] The GeneXpert IV module system is used at Battor.

[b] The GeneXpert HPV assay yields results typically within 58 min (~1 h). Thus, GeneXpert systems in modules 1, 2, 4, 16, 48, and 80 can run 8, 16, 32, 128, 384, and 640 tests in an 8 h shift.

self-sampling for MA-6000 testing than for the other tests. Second, the technique of sampling would influence HPV detection if too many cells are picked up and released, thereby inhibiting the PCR assay [32]. Patient-specific factors potentially related to the diagnostic yield of HPV PCR assay platforms, including the use of gel lubricants; the presence of vaginal discharge, semen, and spermicide creams; vaginal intercourse, douching, and tampon use; as well as the time of menstrual cycle remain controversial in the literature [32–35]. While the higher detection rates recorded for AmpFire and MA-6000, compared to *care*HPV and GeneXpert might also reflect false positivity due to the PCR-plate format of the two tests, quality control measures were taken to mitigate this. Although both platforms have 96 wells, we run a maximum of 94 tests at a time with at least one positive control and one negative control.

Another factor that rears its head in platform selection, based on our experience, is throughput and sample batching. The GeneXpert platform had a low throughput as only four tests could be run in an hour (the platform could only take four cartridges at a time). As we hitchhiked on the National Tuberculosis Programme that procured the platform for tuberculosis testing, there was limited use of this platform due to cost, as women had to pay from their pockets and many could not afford the GeneXpert HPV DNA testing. COVID-19 testing and others would further compound the number of HPV tests that could be performed with the platform. *care*HPV, AmpFire, and MA-6000 have higher throughputs, running up to 94 samples each in less than 2.5 hours (with positive and negative controls). For *care*HPV, however, the samples had to be batched to avoid wasting reagents. There was no need to batch samples for AmpFire and MA-6000; thus, tests could be run on a single sample or any number of

samples up to 94 (with positive and negative controls). While sample batching would not be a problem for large screening programs with large samples collected, this was a big problem for us, because in our setting, which did not depend on funding, women paid out-of-pocket. It could take several weeks from the time the first woman was screened till 90 women were screened before the samples were run.

There is evidence that the quality of self-collected cervicovaginal samples is similar to that of samples obtained by physicians for the detection of CIN2+ lesions, if PCR-based tests are used [36]. GeneXpert, AmpFire, and MA-6000 are PCR-based tests and so self-sampling is likely to yield the same results with these as for health worker-collected samples. *care*HPV, however, is a low-cost version of the Hybrid Capture 2 technology and has lower sensitivity for self-collected samples; a prior study has shown that vaginal *care*HPV testing has lower sensitivity than cervical *care*HPV [37]. This must be considered in weighing the benefits of increased cervical screening coverage against the risk of missing premalignant lesions with self-sampling.

## Summary

It is difficult to compare the advantages and disadvantages of the different platforms, as what works in one setting may not work in another. For example, the 4 module GeneXpert platform can run only 32 samples in 8 h. This was not a problem for us because women paid from their pockets to get screened. We never had more than 32 women in a day for the GeneXpert. In another setting with a national program where women do not have to pay from their pockets, the GeneXpert 4 module will not be good enough, necessitating a platform with higher throughput. Again, what is a challenge in one setting will not be a challenge in another setting. For example, we faced a challenge with long waiting times (for several weeks) to run the *care*HPV because samples had to be batched. Women had to pay from their pockets so we did not have large numbers (up to 90 samples) to run weekly. The reagents once opened could not be kept for long. This would not be a challenge in another setting with a government funded program where there are large numbers of women (who do not have to pay from their pockets) to get screened. The AmpFire and MA-6000 work similarly [20], so we batched samples and ran them every one to two weeks, generally, using whichever we had reagents for. Although we could get results the same day, we generally did not attempt to run the test on the same day. The exception was when we screened inmates at the Nsawam Medium Security Prison and we took the AmpFire platform into the prison and run the tests the same day there [28]. All the platforms are portable and can be transported on outreaches for screening. We have transported the GeneXpert, *care*HPV, and AmpFire platforms on outreaches in other towns and run the tests on the same day (when we got large numbers for the *care*HPV). GeneXpert has minimal user steps and just requires a single transfer of sample to a cartridge with a pipette. AmpFire requires four pipetting steps for dry brush samples and for a liquid-based medium (ThinPrep), centrifuging to get a pellet followed by three pipetting steps. MA-6000 requires 6 pipetting steps for dry brush samples and for a liquid-based medium (ThinPrep), centrifuging to get a pellet followed by five pipetting steps. *care*HPV requires three washing steps and six pipetting steps. This means that minimal training (in pipetting) is required to run the GeneXpert compared to the other tests with also minimal risk of sample contamination (giving false positive results).

## Strengths and limitations

As a strength, this study is the first to perform a four-way comparison of the clinical 'appropriateness', throughput, ease of use, and diagnostic yield of these four HPV testing platforms. In

addition to sharing our broad experience with these devices having used them over several years, our study cohort was relatively large, enabling us to stratify the results of the attendees with adequate statistical power. Despite these strengths, our study was not without limitations. First, this work was done in the clinical setting of Catholic Hospital, Battor, and was not a funded project. Women paid from their pockets to get screened and had the option of choosing tests depending on their availability and cost. The hr-HPV prevalence rates reported may therefore not represent the rates for the general population as women who could not afford HPV DNA testing were not included. We also acknowledge that due to the nature of our study design, direct comparisons between the testing platforms were not possible as none of the samples included here were run on multiple platforms. While this limits the scope of the study to a descriptive analysis, it is crucial to recognize the value this provides in understanding the individual attributes of each platform. This information is indispensable for healthcare providers and policymakers when considering the adoption of a particular platform based on specific needs and constraints. Again, as is common with retrospective cohort studies of this nature, the completeness of data for some sociodemographic and clinical variables represented minor challenges. To mitigate the issue of missingness, we reviewed the data sources thoroughly and included 'missing' as a category where relevant.

## Conclusions

Here, we present our comparison of four different HPV testing platforms for cervical precancer screening in our facility, Catholic Hospital, Battor, Ghana. We describe the strengths and challenges with the different platforms which we believe will be helpful to centers in low (middle) income countries as they transition into using HPV DNA testing for cervical precancer screening. Given our experiences with the different platforms, we posit that the choice of HPV testing platform for program planning cannot be accomplished with a one-size-fits-all approach. In addition to identifying opportunities to merge HPV DNA testing with established public health programs, factors worth considering are the financial implications of platform acquisition, costs to clients, and throughput depending on how large screening programs are.

## Acknowledgments

The authors acknowledge all workers at the main laboratory of Catholic Hospital, Battor, and the laboratory at the CCPTC as well as the staff at the CCPTC and the Department of Obstetrics and Gynecology, Catholic Hospital, Battor, who contributed in various ways towards the screening and management of these women. The authors also thank the Catholic Hospital, Battor for its support.

## Author Contributions

**Conceptualization:** Kofi Effah, Comfort Mawusi Wormenor, Ethel Tekpor, Joseph Emmanuel Amuah, Isaac Gedzah, Seyram Kemawor, Patrick Kafui Akakpo.

**Data curation:** Kofi Effah, Comfort Mawusi Wormenor, Ethel Tekpor, Joseph Emmanuel Amuah, Nana Owusu M. Essel, Isaac Gedzah, Seyram Kemawor, Benjamin Tetteh Hansen, Bernard Hayford Atuguba, Gifty Belinda Klutsey, Edna Sesenu, Stephen Danyo.

**Formal analysis:** Kofi Effah, Comfort Mawusi Wormenor, Ethel Tekpor, Joseph Emmanuel Amuah, Nana Owusu M. Essel, Isaac Gedzah, Stephen Danyo.

**Investigation:** Kofi Effah, Comfort Mawusi Wormenor, Ethel Tekpor, Benjamin Tetteh Hansen, Bernard Hayford Atuguba, Gifty Belinda Klutsey, Edna Sesenu.

**Methodology:** Kofi Effah, Joseph Emmanuel Amuah.

**Software:** Joseph Emmanuel Amuah.

**Supervision:** Kofi Effah.

**Writing – original draft:** Kofi Effah, Ethel Tekpor, Joseph Emmanuel Amuah, Nana Owusu M. Essel, Isaac Gedzah, Patrick Kafui Akakpo.

**Writing – review & editing:** Kofi Effah, Ethel Tekpor, Joseph Emmanuel Amuah, Nana Owusu M. Essel, Patrick Kafui Akakpo.

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
