## [Decision Letter · Decision Letter 0]

12 Jun 2023

PGPH-D-22-02127

Cervical precancer screening using the careHPV, GeneXpert, AmpFire, and MA-6000 HPV systems in Battor, Ghana

Dear Dr. Essel,

Thank you for submitting your manuscript to PLOS Global Public Health. After careful consideration, we feel that it has merit but does not fully meet PLOS Global Public Health’s publication criteria as it currently stands. Therefore, we invite you to submit a revised version of the manuscript that addresses the points raised during the review process.

We look forward to receiving your revised manuscript.

Kind regards,

Natalia M. Rodriguez, PhD, MPH

Academic Editor

Journal Requirements:

1. We do not publish any copyright or trademark symbols that usually accompany proprietary names, eg (R), (C), or TM  (e.g. next to drug or reagent names). Please remove all instances of trademark/copyright symbols throughout the text, including TM on References Section.

2. Some material included in your submission may be copyrighted. According to PLOS’s copyright policy, authors who use figures or other material (e.g., graphics, clipart, maps) from another author or copyright holder must demonstrate or obtain permission to publish this material under the Creative Commons Attribution 4.0 International (CC BY 4.0) License used by PLOS journals. Please closely review the details of PLOS’s copyright requirements here: PLOS Licenses and Copyright. If you need to request permissions from a copyright holder, you may use PLOS's Copyright Content Permission form.

Potential Copyright Issues:

Figs 2 and 3: Please confirm (a) that you are the photographer; or (b) provide written permission from the photographer to publish the photo(s) under our CC-BY 4.0 license.

Reviewers' comments:

Reviewer's Responses to Questions

**Comments to the Author**

1. Does this manuscript meet PLOS Global Public Health’s publication criteria? Is the manuscript technically sound, and do the data support the conclusions? The manuscript must describe methodologically and ethically rigorous research with conclusions that are appropriately drawn based on the data presented.

Reviewer #1: Partly

Reviewer #2: Partly

2. Has the statistical analysis been performed appropriately and rigorously?

Reviewer #1: Yes

Reviewer #2: No

3. Have the authors made all data underlying the findings in their manuscript fully available (please refer to the Data Availability Statement at the start of the manuscript PDF file)?

Reviewer #1: Yes

Reviewer #2: No

4. Is the manuscript presented in an intelligible fashion and written in standard English?

Reviewer #1: Yes

Reviewer #2: Yes

5. Review Comments to the Author

Reviewer #1: This is a well-written and important contribution to the literature to inform HPV test choices for cervical precancer detection. It is very difficult to navigate the technical and practical differences in HPV test platforms, and this study provides a very helpful analysis in a real-world context in Ghana. My main comments relate to providing more information in the methods and results, and considering some additional points for discussion.

Line 67: It would be helpful to add one more sentence here to provide a roadmap for the next four paragraphs (eg, the four HPV DNA tests that are considered in this study are __)

Line 85: With low-throughput machines, in the authors’ opinion, is it logistically feasible to add in HPV testing to existing TB program instruments? Xpert platforms in theory are great to accommodate multi-analyte detection, but I’ve run into issues where the machines are in use full time for TB and HIV testing, which seems to be the case for the authors too (line 300). It might be helpful to add a point into the discussion about whether this was the main factor that limited the overall use of GeneXpert in this study, or if another factor like cost was involved.

Line 92: I disagree that AmpFire is a simple process of sampling *for the user*, especially compared with GeneXpert. I do agree that AmpFire not requiring extraction (if samples are collected directly into Atila buffer) is simpler than requiring extraction like GeneXpert, but I think it’s important to clarify that “simpler” in this case refers to the molecular detection mechanisms, not the user steps. In practice, AmpFire requires a number of user steps that must be carried out by trained personnel to avoid environmental cross-contamination by amplicons due to its PCR plate format. GeneXpert, with minimal user steps that just require transferring sample to a cartridge with a transfer pipette, in my experience, is much simpler.

Line 110: it would be helpful to define what is meant by ‘appropriateness’ earlier, rather than just in the discussion. I would suggest just replacing the term with throughput and sample batching, as defended in the discussion, for clarity.

Line 136: Throughout the whole “Laboratory processing of cervicovaginal samples for HPV DNA assays”, additional details are needed on sample collection. Different platforms are validated with different collection buffers—were all samples collected into ThinPrep? Were any samples converted from one buffer into another? How long were they stored for and under what storage conditions before testing? Were specimens self-collected or physician-collected, and using what sampling swabs/devices? What instruments were used for AmpFire and MA-6000 testing? Some of these are addressed in the discussion and Table 3 but should appear in the methods.

Line 140: “lysate” should be “lysis buffer” I believe

Line 166: “lyse” should be “lysis”

Line 177: what is meant by “preprocessed”? Using what protocol?

Line 203: Were demographics different between the women who were tested by each of the tests?

Table 2: It would be helpful to see more detail in this analysis, and I suggest significantly expanding upon what is currently shown. Besides treatment, what were the percent of positives that correlated to cytologically negative, <cin2, cin2="">

Line 285: Self-sampling has been shown to have higher cell counts, but slightly lower sensitivity than physician-sampling, which is contrasted with self-sampling increasing rates of HPV detection as proposed in the discussion. I would consider the possibility that higher rates of detection may reflect false positivity due to the PCR-plate format of the AmpFire and MA-6000 tests (I’ve seen very high levels of FPs on AmpFire in practice). Was quality control performed to monitor levels of false positivity? If not, I would include in the discussion that this was not evaluated but should be in the future to understand true sample differences versus platform differences.

Were any samples run on more than one platform? How did these compare?

Figure 1: This figure is hard to read with the background chosen. I would recommend cleaning this figure up a bit with a plain background, and if possible, more linear paths through the diagram.</cin2,>

Reviewer #2: Utilizing HPV DNA testing is recommended by the WHO for cervical cancer screening in middle- and low-income countries. Therefore, it is of interest to access and compare performances of various available HPV testing platforms that have been operated in low-resource settings, such as Ghana. The authors of this manuscript aimed to compare the performances of careHPV, GeneXpert, AmpFire, and MA-6000. Operational features (manufacturer, HPV types, sample type, batching, duration, throughput) were demonstrated for each platform. The authors also presented the number of tests performed, number of resulting HR-HPV positive tests and percentages using each platform, and treatments performed on women who had a positive test.

There are several flaws in this study:

1. The authors presented the HR-HPV positivity rates of each platform and attempted to compare the rates between different platforms. However, the rate calculation with respect to each platform was based on a specific cohort of women who received screening using a specific platform. Therefore, the differences in HR-HPV positivity rates between different platforms could potentially be attributed to the differences of the cohorts that the HPV tests were administrated on. The statistical tests and inference on positivity rates between different platforms are not meaningful in this case. Instead, the authors should compare and report the social, demographic, and primary screening details of the four cohorts of women who used different platforms. In addition, the authors should clearly state in the result section that the differences in HPV positivity rates between different platforms may be attributed to the differences of the cohorts.

2. Cytologic and histopathologic diagnoses of the study samples were mentioned in the introduction, however no actual data were reported in the results section. The correlations of HR-HPV testing results and cytologic and histopathologic results should be reported, which would provide some clinical evidence on accuracy of a specific HPV test.

3. The authors presented treatment results and example images of women who underwent HPV testing using different platforms. However, these reports of treatment yield no relevance to the aim of this study and should be deleted from the manuscript.

4. The overall organization, clarity, English writing of the manuscript are suboptimal.

Specific points that need to be addressed:

Title:

• The tile of the manuscript is misleading, as it suggests that this is a study of actual screening for cervical cancer, which it is not. Nor is it a comparison of the four tests for cervical cancer screening, as no samples were tested by more than one of the assays. A more appropriate title would state that this is a description of the operative parameters of four HR-HPV tests among women living in Ghana.

Abstract:

• Line 43, The statement “the limitations of each test in terms of sensitivity and specificity depending on the local HPV genotype prevalence” is incorrect. Sensitivity and specificity are characteristics of a diagnostic test that are not dependent on prevalence of the disease or condition. In fact, sensitivity and specificity were not assessed for the four HPV testing platforms. Therefore, it should not be stated in the abstract.

Introduction

• There is no description of the Cervical Cancer Prevention and Training Center, Battor, where the HPV tests were conducted.

• End of Line 60 what are the aspects the authors refer to? How these aspects are related to the current manuscript.

• Line 71 Digene Hybrid Capture II technology needs reference here.

• Line 73 the authors mentioned the advantages of careHPV. What are the advantages that need to be stated clearly.

• Line 75 careHPV is adequately sensitive. What is the value?

• Line 89-90 how is use of GeneXpert related to reduction of loss of follow-up?

• Line 110-111 the authors should give clear definitions for appropriateness, ease of use, throughput, and diagnostic yield, because they are the four endpoints of this manuscript.

Materials and methods

• A description of CCPTC should be provided. The authors should also provide a brief description of the population who attend CCPTC.

• Figure 1 presents the algorithm for screening and treatment in their institution, which is irrelevant to the aim of this manuscript.

• The authors should also describe the specific cohorts of women who received screening based on specific HPV test platforms.

Results

• Table 1 should be presented by specific cohorts of women who were screened using specific HPV testing platforms.

• Presentations of treatments are irrelevant to the aim of this study

• Table 2 needs footnotes indicating HPV types tested in each assay, for example in careHPV, P3, P4, P5 of GeneXpert, etc.

• Table 3 needs footnotes explaining 8h throughput of GeneXpert.

Discussion and Conclusion

• This is a descriptive report of operational functions of the four HPV testing platforms. Because of the study design (each platform was performed among a specific cohort of women), no actual comparisons were made between the four platforms, except for descriptions of the functional features of each platform. Therefore, the authors should clearly state the scope of the study.

• The authors aimed to compare the four platforms in terms of appropriateness, ease of use, throughput, and diagnostic yield. The discussion section should be organized in discussions of the above features in separate paragraphs.

• The authors should also provide a summary paragraph discussing advantages and disadvantages of each platform in low-resource setting use. In addition, the authors recommendation in use of different platform under various circumstances would be beneficial.

6. PLOS authors have the option to publish the peer review history of their article (what does this mean?). If published, this will include your full peer review and any attached files.

**Do you want your identity to be public for this peer review?** For information about this choice, including consent withdrawal, please see our Privacy Policy.

Reviewer #1: No

Reviewer #2: No

---

## [Editor Report · Decision Letter 1]

8 Aug 2023

Evaluating operational parameters of the careHPV, GeneXpert, AmpFire, and MA-6000 HPV systems for cervical precancer screening: Experience from Battor, Ghana

PGPH-D-22-02127R1

Dear Dr Essel,

We are pleased to inform you that your manuscript 'Evaluating operational parameters of the careHPV, GeneXpert, AmpFire, and MA-6000 HPV systems for cervical precancer screening: Experience from Battor, Ghana' has been provisionally accepted for publication in PLOS Global Public Health.

Best regards,

Natalia M. Rodriguez, PhD, MPH

Academic Editor